# SEARCHING FOR ROBUST POINT CLOUD DISTILLATION

## ABSTRACT

Deep Neural Networks (DNNs) have shown remarkable performance in machine learning; however, their vulnerabilities to adversarial attacks have been exposed, particularly in point cloud data. Neural Architecture Search (NAS) is a technique for discovering new neural architectures with high predictive accuracy, yet its potential for enhancing model robustness against adversarial attacks remains largely unexplored. In this study, we investigate the application of NAS within the framework of knowledge distillation, aiming to generate robust student architectures that inherit resilience from robust teacher models. We introduce RDANAS, an effective NAS method that utilizes cross-layer knowledge distillation from robust teacher models to enhance the robustness of the student model. Unlike previous studies, RDANAS considers the teacher model's outputs and automatically identifies the optimal teacher layer for each student layer during supervision. Experimental results on ModelNet40, ScanObjectNN and ScanNet datasets demonstrate the efficacy of RDANAS, revealing that the neural architectures it generates are compact and possess adversarial robustness, which shows potential in multiple applications.

## 1 INTRODUCTION

Deep learning models, especially deep neural networks (DNNs), have achieved remarkable success in various fields, such as image recognition and natural language processing. Recently, researchers have proposed the technology of NAS, an automated machine learning process that can explore and discover new deep learning model architectures to improve prediction accuracy for specific tasks. In key domains such as computer vision (Chen et al., 2020; Grill et al., 2020; He et al., 2020; Bao et al., 2022; He et al., 2022; Zhou et al., 2022; Zhuang et al., 2021; 2019) and natural language processing (Devlin et al., 2019; Brown et al., 2020), NAS (Real et al., 2017; Zoph & Le, 2017; Tan et al., 2019) has become an effective tool for enhancing model performance. The encoding of architectures into a unified hypernetwork with shared weights has been achieved in recent research (Liu et al., 2019; Cai et al., 2019; Wu et al., 2019; Wan et al., 2020; Nath et al., 2020), leading to a substantial reduction in computational time through the application of gradient descent optimization, particularly in terms of accuracy, parameter count, and computational efficiency. Although NAS has achieved some success in enhancing model performance, these models show significant vulnerability when faced with carefully designed adversarial attacks, and research on its performance under adversarial attacks is still relatively limited. Adversarial attacks can mislead models into making incorrect predictions by adding subtle, imperceptible disturbances to the input data. This vulnerability seriously threatens applications requiring high reliability, such as autonomous driving and medical diagnosis. It remains uncertain whether the architecture derived from search algorithms exhibits robustness, should robustness be present, it is yet to be determined which specific layers and parameters exert a substantial influence, and the feasibility of transferring such robust architectures to other models is also under investigation.

Adversarial attacks are a technique that misleads deep learning models by introducing subtle perturbations to the input data. These perturbations are typically imperceptible to human observers. Still, they are sufficient to cause erroneous predictions in models, which involve adding meticulously designed noise to raw data to mislead machine learning models into making incorrect predictions. The methods for carrying out such attacks are diverse, including the Fast Gradient Sign Method (FGSM) (Goodfellow et al., 2014), Projected Gradient Descent (PGD) (Madry et al., 2018), and the Joint Gradient Based Attack (JGBA) Ma et al. (2020), which primarily identify misleading perturbations by calculating gradients or optimization processes. In addition, there are physical disturbances, such

as adding noise points or adjusting brightness, as well as geometric transformations like scaling and cropping, used to generate adversarial examples. Furthermore, perturbations are designed based on the model's decision boundaries, intended to find the most minor input changes that lead to incorrect predictions. All these attack methods are designed to test the robustness of models, that is, their ability to resist carefully crafted input disturbances, thereby enabling researchers to develop more effective defense strategies to enhance the security and reliability of models.

In the context of adversarial attacks (Szegedy et al., 2014), enhancing the robustness of models becomes particularly important. Robustness refers to the ability of a model to maintain its performance in the face of input disturbances. To improve the robustness of models under adversarial attacks, this paper proposes a new algorithm called RDANAS, which learns from a well-trained, robust teacher model through cross-layer knowledge distillation (KD) to enhance the robustness of the student model. KD (Hinton et al., 2015) is another technique to improve model performance. It transfers knowledge from a large, complex teacher model to a smaller student model to enhance the student model's performance, which reduces the model's computational cost and improves its generalization ability to some extent. The core advantage of the RDANAS algorithm lies in its ability to automatically search for the optimal neural architecture and optimize it under adversarial attacks to enhance the student's robustness. Our method can discover compact and efficient model architectures and maintain high prediction accuracy when facing adversarial attacks. We have conclusively demonstrated that specific robust layers within the teacher model are pivotal for enhancing robustness performance, learning the feature maps from these layers compels the encoder to extract representative feature maps that may elude capture by the student model alone, underscoring the significance of designing search loss functions tailored for distilling student models, which aligns adversarial training with the evolution of attention mechanisms in vision where feature modeling plays a crucial role. To verify the effectiveness of the RDANAS algorithm, we conduct experiments on multiple datasets and demonstrate its performance under different attack methods. The results proved that RDANAS can find neural architectures that are both compact and robust, which is significant for practical applications.

Our contributions are summarized as follows:

- We propose RDANAS, a novel neural architecture search method that strikes an optimal balance between robustness and predictive accuracy through cross-layer knowledge extraction, allowing student models to inherit robustness without specialized robustness training.
- RDANAS innovates by combining neural architecture search with knowledge distillation, introducing a teaching framework that permits learnable connections between teacher and student model layers, thus advancing state-of-the-art neural network design.
- RDANAS discovers compact neural structures with reduced model size and inference cost and significantly improves adversarial robustness, as evidenced by higher clean and PGD accuracy across multiple datasets.

## 2 RELATED WORK

### 2.1 KNOWLEDGE DISTILLATION

KD is a sophisticated model compression technique that enables a compact and efficient student model to emulate a larger, more intricate teacher model, thereby reducing computational demands and equipping the student model with the ability to grasp the nuanced features and decision-making processes inherent in teacher models (Hinton et al., 2015). KD has evolved to include multi-level knowledge distillation (Romero et al., 2015; Yim et al., 2017; Zagoruyko & Komodakis, 2017; Tian et al., 2020; Sun et al., 2019), where insights are extracted from various intermediate layers of the teacher model, allowing the student model to assimilate a broader spectrum of data characteristics. Furthermore, certain KD approaches (Yim et al., 2017), (Zagoruyko & Komodakis, 2017) and (Li et al., 2019) concentrate on transferring attention or feature maps to guide the student model's focus on critical areas of input data, which is particularly beneficial in tasks like image recognition where such maps can direct the model's attention to pivotal image segments, enhancing accuracy. (Dong et al., 2024) Proposed an improved teacher training method that combines two types of regularization. (Sengupta et al., 2024). The proposed RDANAS method takes KD a step further by incorporating cross-layer connections and trainable mappings to bolster the student model's robustness, not only

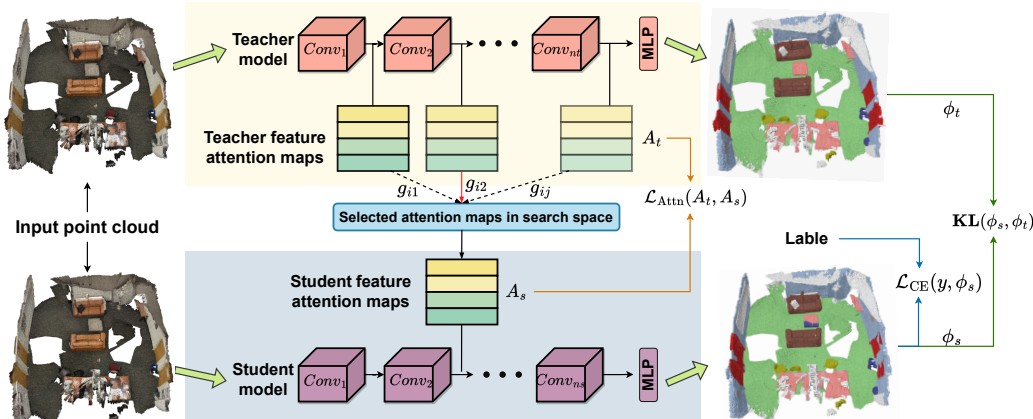

Figure 1: RDANAS training schematic, the robustness of the student model is enhanced by connecting the attention maps of students with the layers of the teacher model. Specifically, the teacher layer that matches the most is sought for each student layer. Here, $g_{ij}$ represents the Gumbel weight between the $i$-th student layer and the $j$-th teacher layer. Additionally, RDANAS constructs an efficient model by searching for the optimal number of filters in each convolutional block of the student model.

facilitating knowledge transfer from the teacher model but also enhancing the student model's resilience to input perturbations, thus providing more excellent stability against adversarial attacks.

## 2.2 NEURAL ARCHITECTURE SEARCH

Neural Architecture Search represents an automated approach to crafting neural network architectures, eliminating the need for manual intervention and systematically probing the architectural landscape to uncover models that excel in specific tasks. While early NAS methodologies hinged on resource-intensive evolutionary algorithms (EA) (Real et al., 2017) and reinforcement learning (RL) (Zoph & Le, 2017; Tan et al., 2019), more recent advancements (Liu et al., 2019; Cai et al., 2019; Wu et al., 2019) have pivoted towards weight-sharing hypernetworks that leverage gradient descent for architectural optimization, significantly curtailing computational expenses and broadening NAS applicability. The quintessential objective of NAS is to identify an architecture that strikes a harmonious balance among accuracy, parameter count, and computational efficiency. This balance is especially crucial in resource-constrained environments. Some NAS techniques (Peng et al., 2020; Nath et al., 2023) have sought to enhance search efficacy and model performance by incorporating KD, harnessing the teacher model's knowledge to expedite the student model's convergence and refine its ultimate performance. (Yue et al., 2022) integrates adversarial training with NAS to enhance machine learning models' accuracy, latency, and robustness simultaneously. Our RDANAS algorithm encapsulates the synergies of NAS and KD, employing cross-layer knowledge distillation to unearth neural architectures that are inherently robust, capable of autonomously identifying the most optimal configurations while ensuring these architectures maintain a heightened level of robustness in the face of adversarial attacks.

## 3 METHOD

This section delves into harnessing the knowledge encapsulated by an efficient teacher model to identify an architecture that is both potent and efficacious. Section 3.1 details the point attention maps, entailing the identification of salient features within the intermediate strata of the teacher model, positing that emulating the focal points of attention can substantially bolster the student model's resilience against adversarial assaults. Sections 3.2 and 3.3 expand on mentor strategies and architectural search algorithms tailored to seek out appropriate teacher layers for the student layers, with our aspiration extending beyond merely augmenting the model's robustness to encompass the discovery of an efficient student architecture. Section 3.4 elucidates the optimization objectives integral to both the search and training processes, and although RDANAS is adept at uncovering

student architectures with inherent resistance to adversarial incursions, it is further augmented through adversarial training to bolster the student model's robustness.

## 3.1 POINT ATTENTION MAPS

The introduction of point attention maps in our methodology is an intuitive illustration for comprehending how network layers focus on different regions of the input data. Expressly, point attention maps are generated from activation tensors, where the activation tensor $A \in \mathbb{R}^{C \times N}$ represents the output of the convolutional layer, and $C$, $N$ denotes the number of channels and spatial dimensions, respectively. By defining a mapping function $\mathcal{F}: \mathbb{R}^{C \times N} \to \mathbb{R}^{D \times N}$, we can transform the activation tensor $A$ into an attention map $\mathcal{F}(A) \in \mathbb{R}^{D \times N}$, typically computed based on the element-wise squared sum of the activation tensor, that is, $[\mathcal{F}(A)]_{c,n} = \sum_{c=1}^{C} A_{c,n}^2$. In our RDANAS, point attention maps not only enhance the interpretability of the network but also become crucial for cross-layer learning. The point attention maps of the student layer are compared and learned from those of the teacher layer to emulate how the teacher layer focuses on the input data. To achieve this process, all point attention maps need to be interpolated to a typical dimension for effective comparison and learning. The RDANAS searches for the optimal teacher layer for each student layer so that the point attention maps of the student layer are similar to those of the mentor layer, thereby learning how to focus on the critical parts of the input data. Furthermore, RDANAS introduces an attention loss function, which quantifies the differences between the point attention maps of the student and teacher layers and minimizes this loss during the training process, thereby guiding the learning of the student layer.

## 3.2 TEACHER DISTILLATION SEARCH

The teacher distillation search is a pivotal step designed to identify the most suitable teacher layer for each student layer to guide the learning of attention and feature representation. This process involves searching through various layers of the teacher model to ascertain which layer is best suited to serve as a teacher for the student, resulting in a potentially vast search space, especially when the number of layers in the student and teacher models is large. To conduct the teacher distillation search computationally efficiently, we employed the Gumbel-Softmax (Jang et al., 2017). This reparameterization trick can be considered a differentiable approximation of the arg max function, which is defined as $g(v) = [g_1, \ldots, g_n]$, $g_i = \frac{\exp((w+\epsilon)/\tau)}{\sum_k \exp((w_i+\epsilon_i)/\tau)}$, where $w = [w_1, \ldots, w_n]$ represents the network parameters, $\epsilon_i \sim N(0,1)$, and $\tau$ is the temperature parameter, $g_i$ denotes the outcome of the Gumbel-Softmax function represents the Gumbel noise. In RDANAS, each student layer is associated with multiple Gumbel weights, used during the search process to represent the connection strength between student and teacher layers. As the search process progresses, the temperature parameter of the Gumbel weights gradually decreases, causing the weight encoding to approach a one-hot vector.

Furthermore, we defined an attention loss function to optimize the similarity between the point attention maps of student and teacher layers:

$$\mathcal{L}_{Attn}(A_s, A_t) = \frac{1}{n_s \cdot n_t} \sum_{i=1}^{n_s} \sum_{j=1}^{n_t} g_{ij} \left\| \mathcal{F}(A_{s,i}) - \mathcal{F}(A_{t,j}) \right\|_2^2. \tag{1}$$

This loss function utilizes Gumbel weights $g_{ij}$ to weight the differences between the activation tensors of student $A_s$ and teacher $A_t$ layers, $\| \cdot \|_2$ is the $\ell_2$ norm. During the search phase, the model's Gumbel weights $g_{ij}$ are affected by the temperature $\tau$. Other parameters are updated through the RDANAS search loss function, including optimizing the student layer's architectural weights and determining the optimal teacher layer for each student layer via Gumbel weights. Once the search phase is completed, each student layer will select the teacher layer with the most potent connection as its teacher. Thus, the student model can learn from the teacher model how to focus on the critical parts of the input data, thereby enhancing its adversarial robustness. Teacher distillation search is one of the essential steps in the proposed RDANAS method, enabling the student model to inherit robustness from the teacher model while maintaining the model's compactness and efficiency. RDANAS can identify compact neural network architectures that perform well under adversarial attacks through teacher distillation search.

### 3.3 ARCHITECTURAL SEARCH

We introduce an innovative approach in the domain of neural network architecture search, particularly in seeking efficient architectures with low latency for student models. Our method aims to identify a student model architecture with low latency. We define a set of potential filter counts $H = \{h_1, h_2, \ldots, h_n\}$ and establish the weighted sum of all possible outputs for each convolutional block's output $Z$, wherein the weights are determined by the Gumbel weights $g(i)_w$, such that $Z = \sum_{i=1}^{n} g(i)_w z_i$. To assess the computational efficiency of each filter choice, we define the number of FLOPs (floating point operations per second) $f(i)$ and calculate the total FLOPs for the convolutional block, which is the sum of the products of all Gumbel weights and their corresponding FLOPs, that is, $\sum_{i=1}^{n} g(i)_w f(i)$. The optimization method employs Stochastic Gradient Descent (SGD) differentially. The Gumbel weights are optimized via an exponentially decaying temperature parameter, causing the weights to approximate a one-hot encoded vector, where most weights are close to 0, and only one weight is close to 1. In the discovered architecture, the number of filters in the convolutional block is determined by the filter choice with the largest Gumbel weight. Figure 1 illustrates the architecture search process of RDANAS, providing an intuitive perspective for understanding the entire search strategy.

### 3.4 RDANAS LOSS

Building upon the current state-of-the-art NAS techniques, the framework encompasses two principal stages: search and training. During the search phase, we focus on periodically updated Gumbel weights that are pivotal in the student-teacher network connections and the filter selection discussed. Specifically, the Gumbel weights we update include the weights for student-teacher connections $g_{ij}$ and the weights for filter selection $g(i)_w$. These weights are optimized through a specifically designed search loss function, which will be elaborated upon in this section. The loss functions utilized in our algorithm to guide neural architecture search and model training are introduced. The following is a summary of this section.

**Search Phase** The search loss function integrates multiple components, including cross-entropy loss, KL divergence, attention loss, and latency penalty term. Cross-entropy loss is employed to measure the discrepancy between the student model's predictions and the true labels. At the same time, the KL divergence is utilized to measure the divergence between the student model's output probability distribution and the teacher model's output probability distribution. The latency penalty term encourages the discovery of architectures with lower computational latency, taking into account the model's FLOPs to ensure the model's efficiency. The general form of the search loss function is given by:

$$\mathcal{L}(y, \phi_s, \phi_t, A_t, A_s) = -y \log \phi_s + KL(\phi_s, \phi_t) + \gamma_s \mathcal{L}_{Attn}(A_t, A_s) + n_f(G), \tag{2}$$

where $y$ represents the one-hot encoding of the true labels, $\phi_s$ and $\phi_t$ are the output probabilities of the student and teacher models, respectively, $KL(\phi_s, \phi_t)$ is the Kullback-Leibler divergence, $\mathcal{L}_{Attn}(A_t, A_s)$ is the attention loss, $\gamma_s$ is the weight coefficient for the attention loss, and $n_f(G)$ is the latency penalty term, The calculation of $n_f(G)$ is as follows:

$$n_f(G) = \sum_{\ell=1}^{L} m_\ell \sum_{i=1}^{n_\ell} g(i)_w^\ell \cdot f(i)^\ell, \tag{3}$$

where $L$ represents the number of layers in the network, $m_\ell$ is the total number of filter choices in layer $\ell$, $n_\ell$ denotes the number of filter options in layer $\ell$, $g(i)_w^\ell$ is the Gumbel weight corresponding to the $i$-th filter choice in layer $\ell$, and $f(i)^\ell$ represents the number of FLOPs corresponding to the $i$-th filter choice in layer $\ell$. Where $G$ is the vector of Gumbel weights.

**Train Phase** Following the search phase, the selected architecture will be trained using the training loss function, which is as follows:

$$\mathcal{L}(y, \phi_s, \phi_t, A_t, A_s) = \mathcal{L}_{\text{CE}}(y, \phi_s) + KL(\phi_s, \phi_t) + \gamma_t \mathcal{L}_{\text{Attn}}(A_t, A_s), \tag{4}$$

where $\mathcal{L}_{\text{CE}}(y, \phi_s)$ is the cross-entropy loss, and $\gamma_t$ is an additional normalization constant used to balance the attention loss. Upon completion of the search phase, the selected architecture will be trained using the RDANAS training loss function defined above.

It's worth noting that, RDANAS comprises cross-entropy loss and attention loss and may also include a loss term for adversarial training. RDANAS algorithm permits the integration of adversarial training techniques, such as TRADES (Zhang et al., 2019), with the search and training losses to further enhance the model's robustness. Through these loss functions, the algorithm can effectively optimize the model's accuracy and robustness during the search and training phases while maintaining the computational efficiency of the model.

# 4 EXPERIMENTS AND ANALYSIS

We designed a comprehensive set of comparative and ablation experiments on three datasets to evaluate the effectiveness of our approach rigorously.

## 4.1 IMPLEMENTATION DETAILS

**Datasets** We conducted experiments on ModelNet40 (Wu et al., 2015), ScanObjectNN (Uy et al., 2019) and ScanNet (Dai et al., 2017). For details of the dataset, please refer to Appendix A.

**Training details** Across various datasets, we initiate the process with a search phase, during which we utilize RDANAS to train the model to optimize the loss function and determine the channel count and associated teacher layer for each student layer. Experiments are executed with a diverse array of search spaces and a spectrum of robust teacher models, ensuring a comprehensive evaluation of the model's performance. Throughout this section, our model is denoted as RDANAS-T, wherein 'T' indicates the different robust teacher model. We employed three distinct robust teacher models: PointNetWang et al. (2019), DGCNN Wang et al. (2019), and PointNext (Qian et al., 2022), designated as P, D, and PX, respectively. For instance, RDANAS-PX denotes the model trained with PointNext as the adversarial robust teacher, showcasing the specificity and adaptability of our approach. In the training of RDANAS, we utilized a CrossEntropy loss function with label smoothing, the AdamW optimizer, an initial learning rate of 0.001, weight decay set to $10^{-4}$, a cosine decay schedule, a batch size of 32, and an RTX 4090 GPU. We annealed the training process to zero in accordance with a cosine decay schedule. After a 200-epoch search phase, the identified architecture was retrained from the ground up for an additional 200 epochs, employing the training loss. The temperature parameter ($\tau$) within the Gumbel-Softmax was initialized to 5.0 and was exponentially annealed by a factor of $e^{-0.045}$ per epoch throughout the search phase. The hyperparameters $\lambda_s$ and $\lambda_t$ were uniformly set to 1. in all experiments0. Throughout the search phase, 80% of the data in each batch was allocated to optimize the model weights, while the remaining 20% was dedicated to the optimization of the architecture weights. For the robustness evaluation, we identified three potent attack vectors: FGSM (Goodfellow et al., 2014), PGD (Madry et al., 2018), and JGBA Ma et al. (2020). The adversarial perturbations were assessed under the $L_\infty$ norm, with the magnitude of perturbations capped at 8/255 (equivalent to 0.031).

## 4.2 COMPARISON WITH STATE-OF-THE-ART

In this section, we undertake a comparative analysis of the robustness of our method against other state-of-the-art efficient robust models. Including the effectiveness of different variants of our method.

**Efficient and Robust Models** Table 1 and Table 2 present a comparative evaluation of our RDANAS against other SOTA models, highlighting the distinct advantages of our approach. Each instance of the RDANAS model was meticulously trained utilizing robust teacher models, ensuring a solid foundation for performance enhancement. The results of the Tables show that RDANAS markedly surpasses all other models in terms of accuracy, underscoring the effectiveness of our training strategy. Notably, our RDANAS models exhibit a significant reduction in size, which is an encouraging outcome, indicating the efficiency of our model architecture. For example, RDANAS has achieved an enhancement in accuracy of over 10% when compared to most other models of comparable size on the ScanObjectNN dataset, demonstrating the superior robustness of our method. Furthermore, when juxtaposed with models exhibiting similar accuracy, our model reflects a reduction in size by more than 10%, showcasing the balance between performance and compactness that is crucial for practical applications, as shown in Table 4. These findings underscore the potential of RDANAS as a leading candidate for efficient, robust models.

Table 1: Comparison with clean dataset. The average result of 3 runs is given in brackets.

| Method | ScanObjectNN | | | ModelNet40 | |
|---|---|---|---|---|---|
| | OBJ-BG | OBJ-ONLY | PB-T50-RS | 1k P | 8k P |
| PointNet Qi et al. (2017a) | 73.3 | 79.2 | 68.0 | 89.2 | 90.8 |
| PointNet++ Qi et al. (2017b) | 82.3 | 84.3 | 77.9 | 90.7 | 91.9 |
| PointCNN Li et al. (2018) | 86.1 | 85.5 | 78.5 | 92.2 | - |
| DGCNN Wang et al. (2019) | 82.8 | 86.2 | 78.1 | 92.9 | |
| MinkowskiNet Choy et al. (2019) | 84.1 | 86.1 | 80.1 | - | - |
| PointTransformer Zhao et al. (2021) | - | - | - | 93.7 | - |
| PointMLP Ma et al. (2022) | 88.7 | 88.2 | 85.4 | **94.5** | - |
| SimpleView Lai et al. (2022) | - | - | 80.5±0.3 | 93.9 | - |
| PointNeXt Qian et al. (2022) | 91.9 | 91.0 | 88.1 | 94.0 | - |
| RDANAS (PointNet) | 78.7(78.6) | 82.2(81.6) | 72.7(72.5) | 90.3(90.1) | 90.9(90.8) |
| RDANAS (DGCNN) | 85.3(85.2) | 88.2(87.9) | 80.5(80.4) | 92.8(92.7) | 83.5(83.5) |
| RDANAS (PointNeXt) | **92.3**(92.2) | **91.3**(91.2) | **88.5**(88.3) | 94.4(94.3) | **94.5**(94.5) |

Table 2: Test accuracy comparison (higher is better) on the full test datasets, where "-" indicates no result.

| | | PointNet | PointNet w/ DUP-Net Zhou et al. (2019) | PointNet w/ IF-Defense Wu et al. (2020) | PointNet w/ RPL Gould et al. (2021) | RDANAS w/ PointNet | RDANAS w/ DGCNN | RDANAS w/ PointNeXt |
|---|---|---|---|---|---|---|---|---|
| | No attack | **90.15** | 89.30 | 87.60 | 84.76 | 89.76 | 89.96 | 89.76 |
| ModelNet40 | FGSM Goodfellow et al. (2014) | 45.99 | 61.63 | 38.75 | 0.04 | 53.65 | 58.92 | **67.39** |
| | JGBA Ma et al. (2020) | 0.00 | 1.14 | 5.37 | 0.00 | 18.65 | 19.25 | **29.51** |
| | No attack | **84.61** | 83.62 | 80.19 | 76.02 | 81.87 | 80.52 | 83.64 |
| ScanNet | FGSM Goodfellow et al. (2014) | 45.66 | 73.67 | 71.14 | 1.70 | 68.32 | 71.34 | **78.62** |
| | JGBA Ma et al. (2020) | 0.00 | 7.77 | 13.45 | 0.00 | 18.45 | 21.43 | **33.16** |

**Various Perturbation Budgets** To substantiate the efficacy of RDANAS, we conducted a comparative analysis with existing defense mechanisms across a spectrum of perturbation budgets. Figure 2 illustrates a comparative analysis of various methods in the context of JGBA and FGSM adversarial attacks. It reveals that RDANAS demonstrates superior performance across the board relative to its peers under all considered perturbations. With an escalation in perturbation magnitude, RDANAS exhibits a markedly enhanced performance compared to alternative approaches, highlighting its resilience against adversarial perturbations. Specifically, at a perturbation magnitude of 0.1, RDANAS achieves an approximate 20% improvement over other methods in the context of both JGBA and FGSM attacks, underscoring its robustness and effectiveness in real-world scenarios where such perturbations are likely to be encountered. These findings underscore the potential of RDANAS as a leading candidate in adversarial defense mechanisms.

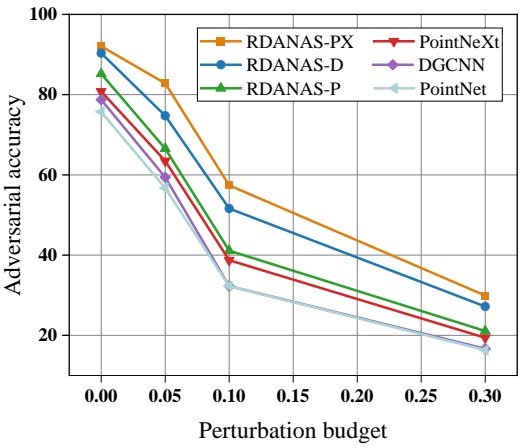

Figure 2: The results of various models' adversarial accuracy under various perturbation budgets on the ModelNet40 dataset.

**Compare with Defense Strategies** Additionally, a comparative analysis was conducted between our method and a spectrum of Defense Strategies. The Robust PointNeXt was employed as the teacher model across all knowledge distillation approaches, with the training of three distinct student architectures: PointNet (Qi et al., 2017a), PointNet++ (Qi et al., 2017b), and DGCNN Wang et al. (2019). As depicted in Table 3, models trained employing our paradigm are distinctly positioned in the upper right quadrant of the graph, underscoring the efficacy of the intermediate cross connections. The performance of RDANAS architectures, when trained with IF-Defense, closely mirrors that of models trained via our method. Furthermore, all RDANAS-trained models markedly surpass

other methodologies in terms of both clean and adversarial accuracy. These findings highlight the robustness and superiority of our approach in the context of defense.

Table 3: Leaderboard. Blue: best in a row. Gray: worst in a row. Compared with existing augmentation methods, our method achieves the SOTA performance.

| Defense & (Acc) | Model | Clean | PGD | Drop | KNN | Add | IFGM | Perturb |
|---|---|---|---|---|---|---|---|---|
| Ours **(80.62)** | PointNet | 88.47 | 77.64 | 82.36 | 86.43 | 87.28 | 85.87 | 88.48 |
| | PointNet++ | 88.89 | 74.41 | 84.27 | 86.87 | 87.26 | 88.17 | 88.60 |
| | DGCNN | 88.73 | 76.83 | 81.68 | 87.47 | 88.25 | 87.76 | 87.46 |
| IF-Defense (Wu et al., 2020) (78.4) | PointNet | 85.33 | 44.89 | 65.19 | 82.46 | 85.41 | 82.86 | 85.01 |
| | PointNet++ | 87.52 | 38.61 | 73.01 | 85.56 | 87.20 | 85.37 | 87.12 |
| | DGCNN | 87.88 | 40.32 | 71.48 | 85.78 | 86.75 | 85.86 | 87.32 |
| SOR (Zhou et al., 2019) (75.19) | PointNet | 86.95 | 42.10 | 57.86 | 80.06 | 86.10 | 84.16 | 85.53 |
| | PointNet++ | 88.57 | 25.00 | 66.25 | 85.13 | 88.70 | 87.72 | 88.98 |
| | DGCNN | 88.57 | 18.00 | 66.94 | 85.25 | 87.88 | 87.64 | 87.44 |
| No Defense (67.06) | PointNet | 87.64 | 34.32 | 59.64 | 45.10 | 71.76 | 74.59 | 85.58 |
| | PointNet++ | 89.30 | 15.56 | 71.47 | 54.25 | 72.37 | 81.22 | 88.17 |
| | DGCNN | 89.38 | 18.96 | 73.10 | 70.10 | 83.71 | 86.91 | 88.74 |

**Training Time Budget** A comparison of training time was conducted between our proposed RDANAS and the previously established state-of-the-art robust and efficient methods. As shown in Table 4. The comparative analysis reveals that RDANAS exhibits superior adversarial accuracy compared to its competitors and requires significantly less training time than the majority of baseline methods. In contrast to PointNet, RDANAS demonstrates a marginally longer training duration; nonetheless, it outperforms PointNet with respect to adversarial accuracy and boasts a more streamlined parameter and FLOPs count. These findings underscore the potential of RDANAS as a leading approach in the field of adversarial learning, offering a promising balance between efficiency and robustness.

Table 4: Performance of various efficient and robust methods on ModelNet-40 dataset, $*$ denote 20 steps attack.

| Method | Clean | $PGD^{20*}$ | # Params (M) | FLOPs (G) | Training Time (h) |
|---|---|---|---|---|---|
| PointNet (Qi et al., 2017a) | 82.93 | 29.79 | 3.5 | 0.9 | 8 |
| DGCNN (Wang et al., 2019) | 92.91 | 27.67 | 1.8 | 4.8 | 24 |
| RDANAS-P | 90.32 | 32.67 | 3.0 | 0.7 | 14 |
| RDANAS-D | 92.81 | 33.46 | 1.3 | 3.6 | 21 |
| RDANAS-PX | **94.40** | 48.28 | 1.2 | 1.4 | 12 |
| RDANAS-P + TRADES | 88.94 | 35.76 | 3.1 | 0.7 | 8 |
| RDANAS-D + TRADES | 89.95 | 38.79 | 1.3 | 3.6 | 21 |
| RDANAS-PX + TRADES | 89.42 | **50.24** | 1.3 | 1.4 | 14 |

## 4.3 ABLATION STUDY

A comparative analysis of three distinct training paradigms was implanted to demonstrate the significance of teacher-student cross-layer connections within RDANAS. The search and training procedures were executed in the initial paradigm utilizing cross-entropy loss without a teacher model, termed the standard approach. The subsequent approach involved searching and training by minimizing cross-entropy loss and standard KL divergence with a robust teacher model. Lastly, the third paradigm encompasses RDANAS, integrating all three components: cross-entropy loss, KL divergence, and cross-layer student-teacher connections. As presented in Table 5, a comparison of the performance of RDANAS's components was conducted. It was observed that, in comparison to the standard network, both KL and ICC (intermediate cross connections) markedly enhanced robustness across the two training schemes. Ultimately, the amalgamation of KL and ICC, namely RDANAS, surpasses the other methodologies.

Table 5: CE indicates training the student model using the cross-entropy loss function. CE+KL indicates training the student model by minimizing both the cross-entropy loss and the standard KL divergence, in conjunction with a robust teacher model. CE+ICC represents a model trained by minimizing the cross-entropy loss and the intermediate cross-layer connections (ICC).

| Method | CE | KL | ICC | Clean | PGD |
|---|---|---|---|---|---|
| Without adversarial training | ✓ | | | 94.42 | 67.3 |
| Without adversarial training | ✓ | ✓ | | **93.76** | 72.3 |
| Without adversarial training | ✓ | | ✓ | 93.43 | 72.54 |
| Without adversarial training | ✓ | ✓ | ✓ | 93.62 | 76.24 |
| With adversarial training | ✓ | | | 86.85 | 82.67 |
| With adversarial training | ✓ | ✓ | | 88.27 | 84.63 |
| With adversarial training | ✓ | | ✓ | 87.23 | **84.73** |
| With adversarial training | ✓ | ✓ | ✓ | 88.06 | 84.52 |

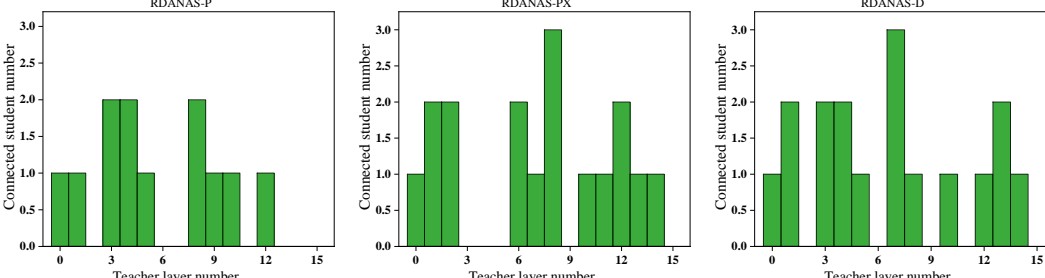

Figure 3: In the RDANAS framework, the depiction of connections between each teacher layer and the corresponding student layers for various student models on the ModelNet40 dataset is presented. The robust teacher model, which is adversarially trained PointNeXt, is selected to be paired with all three student models, resulting in a distinct graph for each student model.

## 4.4 ROBUST CROSS-LAYER CONNECTION

We evaluate the robustness transfer effect within the teacher-student model framework, assuming that extensive teacher training on different datasets leads to varying degrees of robustness across its layers. As a result, specific layers become more resilient and are better suited to enhance the robustness of the student model. RDANAS identifies and leverages these resilient teacher layers to guide the student network. In RDANAS, each student layer is directly associated with different teacher layers. Figure 3 visualizes the connections between the teacher and student layers, highlighting that the 7th and 8th layers of the robust teacher model are particularly influential, indicating that more intermediate connections are established with the student model. More experimental results in Appendix B.

## 5 CONCLUSION

We proposed the RDANAS algorithm, through cross-layer knowledge distillation technology, to successfully extract key knowledge from robust teacher models, significantly enhancing the robustness of student models against adversarial attacks while finding a good balance between computational efficiency and accuracy. Unlike traditional adversarial training methods, RDANAS can improve the robustness of models without undergoing adversarial training, simplifying the model training process. Moreover, the model architectures searched by RDANAS are compact, with fewer parameters and lower computational loads, making them suitable for deployment in resource-constrained environments and easy to maintain and update. The RDANAS algorithm has been extensively tested on multiple standard datasets, proving its wide applicability and strong generalization capabilities. RDANAS has significantly improved adversarial accuracy compared to existing efficient and robust models, demonstrating the potential of cross-layer knowledge distillation in enhancing model robustness. This method emphasizes the effectiveness and practicality of RDANAS in automatically searching for robust and efficient neural architectures, providing a solid foundation for the future deployment of deep learning models in high-risk application fields.

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

## A    TRAINING STRATEGY COMPARISON ON DIFFERENT DATASET

In this section, we provide a detailed summary of the training strategies employed across various versions of our student models, including PointNet (Qi et al., 2017a), DGCNN (Wang et al., 2019), PointNeXt (Qian et al., 2022), and RDANAS, as demonstrated on ModelNet40 (Wu et al., 2015) dataset in Table 6, on ScanObjectNN (Uy et al., 2019) dataset in Table 7, and ScanNet (Dai et al., 2017) dataset in Table 8, respectively. ModelNet40 is a dataset composed of 40 categories, encompassing 12,311 3D CAD object models. These models are classified into 40 distinct categories, among which 9,843 objects are utilized for the training phase, and the remaining 2,468 objects are employed for the testing phase. In our experiments, the surface of each object is uniformly sampled to yield 1,024 points. These point cloud data represent the geometric shape of the objects. Furthermore, the sampled point clouds are rescaled to be within the unit sphere to ensure their uniformity in proportion and scale. ScanObjectNN contains approximately 15,000 scanned real-world objects across 15 different categories. The dataset covers 2,902 unique instances, with each object annotated with global and local coordinates, normal, color attributes, and semantic labels. ScanNet, conversely, is a more complex dataset that comprises 1,513 RGB-D scans covering 707 real indoor scenes. These scenes offer abundant 3D information, including color and depth data, totaling 2.5 million views. Experimentally, 12,445 training point clouds and 3,528 testing point clouds were generated from 17 categories, each containing 1,024 points.

Table 6: The training strategies implemented by diverse methodologies for ModelNet40 classification.

| Method | PointNet | DGCNN | PointNeXt | RDANAS (Ours) |
|---|---|---|---|---|
| Epochs | 200 | 200 | 300 | 200 |
| Batch size | 16 | 16 | 32 | 32 |
| Optimizer | Adam | SGD | AdamW | AdamW |
| LR | $3 \times 10^{-3}$ | $1 \times 10^{-2}$ | $1 \times 10^{-3}$ | 0.001 |
| LR decay | step | step | multi step | multi step |
| Weight decay | 0.0 | $10^{-3}$ | $10^{-4}$ | $10^{-4}$ |
| Label smoothing | ✗ | ✗ | ✗ | ✗ |
| Random rotation | ✗ | ✗ | ✓ | ✓ |
| Random scaling | ✗ | [0.9,1.1] | [0.8,1.2] | [0.8,1.2] |
| Random translation | ✗ | ✗ | ✗ | ✗ |
| Random jittering | ✗ | 0.001 | 0.001 | 0.001 |
| Normal Drop | ✗ | ✗ | ✓ | ✓ |
| Height appending | ✗ | ✓ | ✓ | ✓ |

Table 7: The training strategies implemented by diverse methodologies on the ScanObjectNN dataset.

| Method | PointNet | DGCNN | PointNeXt | RDANAS(Ours) |
|---|---|---|---|---|
| Epochs | 250 | 200 | 250 | 200 |
| Batch size | 32 | 32 | 32 | 32 |
| Optimizer | Adam | SGD | AdamW | AdamW |
| LR | $1 \times 10^{-3}$ | 0.01 | $2 \times 10^{-3}$ | $2 \times 10^{-3}$ |
| LR decay | step | cosine | cosine | cosine |
| Weight decay | $10^{-4}$ | $10^{-4}$ | 0.05 | 0.05 |
| Label smoothing | 0.2 | 0.2 | 0.3 | 0.3 |
| Point resampling | ✗ | ✗ | ✓ | ✓ |
| Random rotation | ✓ | ✗ | ✓ | ✓ |
| Random scaling | ✗ | ✓ | ✓ | ✓ |
| Random translation | ✗ | ✓ | ✗ | ✗ |
| Random jittering | ✓ | ✗ | ✗ | ✗ |
| Height appending | ✗ | ✗ | ✓ | ✓ |

Table 8: The training strategies implemented by diverse methodologies for ScanNet segmentation.

| Method | PointNet | DGCNN | PointNeXt | RDANAS (Ours) |
|---|---|---|---|---|
| Epochs | 200 | 200 | 200 | 100 |
| Batch size | 10 | 16 | 32 | 2 |
| Optimizer | SGD | SGD | Adam | AdamW |
| LR | $1 \times 10^{-2}$ | $5 \times 10^{-1}$ | $1 \times 10^{-3}$ | $1 \times 10^{-3}$ |
| LR decay | step | multi step | multi step | multi step |
| Weight decay | $10^{-3}$ | $10^{-4}$ | $10^{-4}$ | $10^{-4}$ |
| Entire scene as input | ✗ | ✓ | ✗ | ✓ |
| Random rotation | ✓ | ✗ | ✓ | ✓ |
| Random scaling | [0.9,1.1] | [0.9,1.1] | [0.8,1.2] | [0.8,1.2] |
| Random translation | ✗ | ✗ | ✗ | ✗ |
| Random jittering | 0.001 | ✗ | ✗ | ✗ |
| Height appending | ✓ | ✗ | ✓ | ✓ |
| Color drop | ✗ | ✗ | 0.2 | 0.2 |
| Color auto-contrast | ✗ | ✓ | ✓ | ✓ |
| Color jittering | ✗ | ✓ | ✗ | ✗ |

## B    MORE EXPERIMENTAL RESULTS

We conducted a comparative analysis of various RDANAS model versions, specifically evaluating their performance under PGD attacks with differing perturbation budgets on the ScanObjectNN dataset. Figure 4 presents the experimental results with temperature coefficients ($\tau$) set to 5.0, 3.0, and 1.0, arranged from left to right. Following Figure 3, Figure 5 illustrates the diverse depiction of connections between each teacher layer and the corresponding student layers for various student models on the ScanObjectNN dataset.

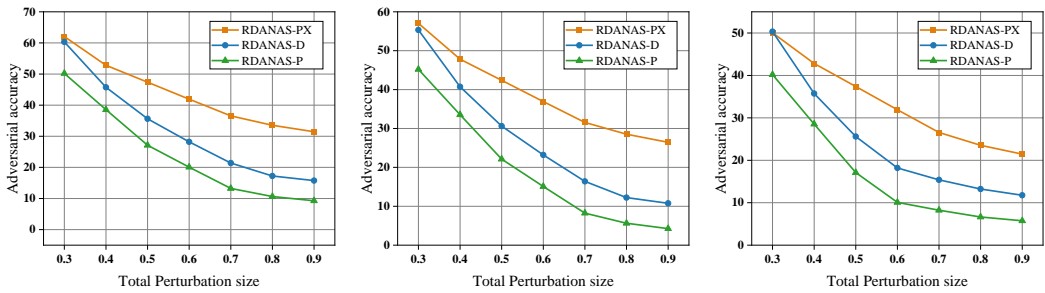

Figure 4: The adversarial accuracy performance of various models on the ScanObjectNN dataset was evaluated across a range of perturbation budgets.

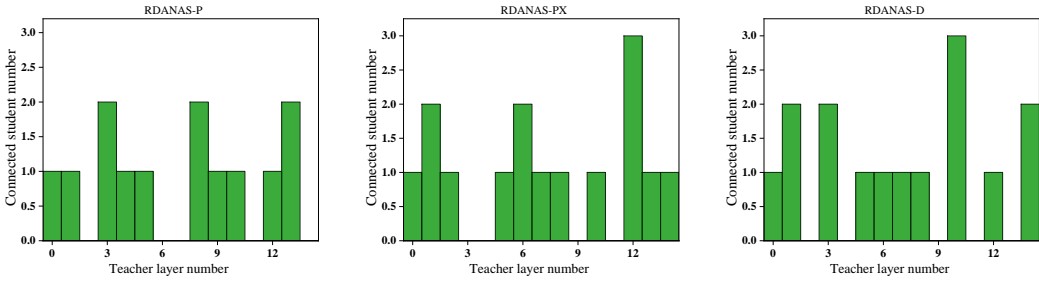

Figure 5: In the RDANAS framework, the depiction of connections between each teacher layer and the corresponding student layers for various student models on the ScanObjectNN dataset is presented.

