# OpenReview forum: "Searching For Robust Point Cloud Distillation"
_ICLR.cc/2025/Conference — ICLR 2025 Conference Withdrawn Submission_

### Official Review · Reviewer_t4WG · 2024-10-23

**Soundness:** 2
**Presentation:** 2
**Contribution:** 2
**Rating:** 3
**Confidence:** 4

**Summary:**

This study explores the use of Neural Architecture Search (NAS) combined with knowledge distillation to improve adversarial robustness in deep neural networks, especially for point cloud data. The proposed method, RDANAS, enhances student model robustness by distilling knowledge from robust teacher models. It optimally selects teacher layers for each student layer, showing improved performance and compact architectures in various datasets, including ModelNet40, ScanObjectNN, and ScanNet. RDANAS demonstrates significant potential for applications requiring robust neural models.

**Strengths:**

1.Detailed background introduction. The authors have introduced the background information in detail, making readers easy to understand.

2.Nice figures. The Figure 1 is nice to read, I appreciate the figure presentations in this paper.

**Weaknesses:**

1.Lack of intuition. I do not see the intuition behind proposed scheme. Since the NAS with knowledge distillation has been studied in previous works [1-3], I am curious about the novelty and intuition of the proposed scheme. I wonder if the proposed approach was directly inspired by [1-3], which mainly focus on  2D image data, and this paper transfers their ideas to 3D point cloud data? In addition, I do not find any scheme design considerations for 3D point cloud data, this is also the reason why I wonder the intuition behind the proposed scheme.

2.Lack of references about defense schemes against 3D point cloud adversarial attacks. In the related work section, the authors just mentioned the NAS and knowledge distillation techniques, however, it is more important to discuss the baseline works that aim to improve point cloud DNNs' robustness because this is the key spotlight of this paper. From my perspective, the authors do not grasp the key point that they need to express in this paper.

3.Old baselines. In the experimental part, the 3D adversarial attacks evaluated are all old baselines, which may not verify the effectiveness of the proposed defense scheme. I encourage the authors to add more recent adversarial attacks in 3D point cloud to support the claim in this paper, such as [4-5].


[1] Block-wisely supervised neural architecture search with knowledge distillation. CVPR 2020

[2] Performance-aware mutual knowledge distillation for improving neural architecture search. CVPR 2022

[3] Multi-fidelity neural architecture search with knowledge distillation.  IEEE Access 2023

[4] Generating transferable 3d adversarial point cloud via random perturbation factorization. AAAI 2023

[5] Curvature-Invariant Adversarial Attacks for 3D Point Clouds. AAAI 2024

**Questions:**

In table 2, why do the authors not include the results of PGD attack?

---

### Official Review · Reviewer_aEYv · 2024-10-27

**Soundness:** 2
**Presentation:** 2
**Contribution:** 1
**Rating:** 1
**Confidence:** 5

**Summary:**

The paper proposes RDANAS, a framework combining Neural Architecture Search (NAS) with cross-layer Knowledge Distillation (KD) to enhance model robustness against adversarial attacks.

**Strengths:**

RDANAS combines NAS and knowledge distillation for robustness against adversarial attack.

**Weaknesses:**

RDANAS seems to follow that same approach as RNAS-CL [1] but on point cloud data. All subsections of the method sections seems to be same as the RNAS-CL paper. Hence, it doesn’t seems to have any contribution other than testing RNAS-CL on point cloud data.

[1] Nath, U., Wang, Y., Turaga, P. et al. RNAS-CL: Robust Neural Architecture Search by Cross-Layer Knowledge Distillation. Int J Comput Vis (2024). https://doi.org/10.1007/s11263-024-02133-4

**Questions:**

- How is RDANAS different than RNAS-CL ?
-  What are the main contributions of the paper beyond those already addressed in RNAS-CL?

**Details Of Ethics Concerns:**

The paper appears to be a plagiarized version of the RNAS-CL [1] paper, as the methodology, equations, experimental setup, and observations closely resemble those of RNAS-CL.

[1] Nath, U., Wang, Y., Turaga, P. et al. RNAS-CL: Robust Neural Architecture Search by Cross-Layer Knowledge Distillation. Int J Comput Vis (2024). https://doi.org/10.1007/s11263-024-02133-4

---

### Official Review · Reviewer_9AdX · 2024-11-02

**Soundness:** 3
**Presentation:** 3
**Contribution:** 3
**Rating:** 6
**Confidence:** 3

**Summary:**

This paper proposes RDANAS, a new algorithm for enhancing the robustness of DNNs against adversarial attacks in point cloud data.  The approach is using NAS and cross-layer knowledge distillation from a robust teacher model.  RDANAS automatically identifies the best teacher layer for each student layer during the learning process.  Experimental results show that RDANAS generates compact and robust neural architectures, demonstrating its potential in various applications.

**Strengths:**

RDANAS has some novel design components by combining Neural Architecture Search with cross-layer knowledge distillation to enhance the robustness of DNNs against adversarial attacks in pointcloud.
Some other contributions in the methodology include: the cross-Layer KD allowing the student model to inherit the robustness of the teacher model without requiring specialized robustness training; the automatic teacher layer selection identifying the optimal teacher layer for each student layer during the learning process, removing the need for manual selection and ensures that the student model learns from the most relevant teacher layers.

From the evaluations, it is shown that RDANAS is effective in generating compact and robust neural architectures, outperforming existing methods in terms of both clean and adversarial accuracy, while also producing smaller models.

The paper provides a thorough evaluation of RDANAS on several benchmark datasets (ModelNet40, ScanObjectNN, and ScanNet) and under different attack methods (FGSM, PGD, and JGBA), showing the generalizability and robustness of  RDANAS.

**Weaknesses:**

There is a lack of details on the computational costs of RDANAS.  Table 4 describes the training time and FLOPs, but it's unclear how well this scales given the potential high cost of NAS.
There is an assumption of the pre-trained robust teacher model in order to generate the student model.  It's unclear how this can be realized in practice.

**Questions:**

Can the authors help clarify these limitations pointed out with respect to computational cost details and the assumption of a robust teacher model?

Furthermore, can the authors please also comment on the expected comparision with other defense methods for adversarial robustness?  Why is such comparison not needed here?

It is also useful to understand how the authors think about the role of attention maps for cross-layer KD impacts the robustness.

---

### Official Review · Reviewer_YYa1 · 2024-11-04

**Soundness:** 2
**Presentation:** 2
**Contribution:** 2
**Rating:** 3
**Confidence:** 3

**Summary:**

The paper introduces RDANAS, a novel method for enhancing model robustness against adversarial attacks in point cloud data by integrating Neural Architecture Search with cross-layer Knowledge Distillation. RDANAS employs robust teacher models to guide the design of compact, adversarial-resistant student architectures by matching optimal teacher layers to student layers, facilitating resilience in the student model without the need for adversarial training. The approach optimizes efficiency and robustness through a Gumbel-Softmax-based teacher-student connection search and attention loss minimization to improve feature extraction. Experimental results on ModelNet40, ScanObjectNN, and ScanNet datasets indicate that RDANAS achieves higher adversarial accuracy and efficiency than existing models. This method shows promise for applications needing robust, resource-efficient neural models in high-risk environments.

**Strengths:**

1. Novelty. RDANAS combines NAS with cross-layer KD to automatically identify compact, resource-efficient model architectures, making it well-suited for deployment in resource-constrained environments. To boost robustness, RDANAS strategically aligns teacher and student layers based on their resilience, enabling the student model to focus on the most effective features and strengthening its resistance to adversarial attacks.

2. Experiments. The paper presents a thorough evaluation of RDANAS.
   - Extensive experimental results highlight RDANAS's effectiveness across multiple datasets, including ModelNet40, ScanObjectNN, and ScanNet, as well as against various attack methods (FGSM, PGD, and JGBA).
   - The authors also examine different defense strategies and perform an ablation study on three distinct training paradigms.

**Weaknesses:**

1. Rationale behind the effectiveness. The underlying reasons for RDANAS’s effectiveness in defending against adversarial attacks are not entirely clear. The implementation in Section 3.2, which discusses selecting optimal teacher layers to guide the student model's focus on critical parts of the input data, lacks a detailed explanation and theoretical basis. It is unclear how this focus directly translates to improved adversarial robustness, leaving questions about the precise mechanisms that enhance the student model's resilience.
2. The methods section is somewhat confusing. While the paper primarily aims to address the challenge of transferring robustness to the student model, only Section 3.2 (*Teacher Distillation Search*) explicitly focuses on robustness design. Other sections discuss topics like low latency, which are unrelated to robustness and could detract from the main objective.
3. The paper lacks evaluation under real-world noise disturbances (e.g., corruption) and physical adversarial attacks. Since the goal is to develop a robust model suitable for deployment in resource-constrained environments, it is essential to conduct experiments with real-world attacks to validate the effectiveness of the proposed method in practical scenarios.
4. Clarity. "Table 3, upper right quadrant of the graph" in line 375 is unclear and could be confusing for readers.

**Questions:**

see weakness

---

### Note · Authors · 2024-11-13

I have read and agree with the venue's withdrawal policy on behalf of myself and my co-authors.